# Study on the Efficacy and Safety of Tedizolid in Japanese Patients

**DOI:** 10.3390/antibiotics13121237

**Published:** 2024-12-23

**Authors:** Kazuhiro Ishikawa, Yasumasa Tsuda, Nobuyoshi Mori

**Affiliations:** 1Department of Infectious Diseases, St. Luke’s International Hospital, Tokyo 104-8560, Japan; morinob@luke.ac.jp; 2Department of Pharmacy, St. Luke’s International Hospital, Chuo-ku, Tokyo 104-8560, Japan; tsuyasu@luke.ac.jp

**Keywords:** tedizolid, antibiotics, linezolid, clinical cure, hematological malignancies, platelet count

## Abstract

**Background/Objective**: Tedizolid (TZD), an oxazolidinone, causes fewer adverse events than linezolid (LZD). However, studies on the long-term efficacy and safety of TZD, particularly in patients with hematological malignancies (HMs), remain limited. This study aimed to evaluate the safety of long-term TZD use in Japanese patients, including those with HM. **Methods**: We retrospectively reviewed the medical records of patients aged 15 years and older who received TZD treatment at St. Luke’s International Hospital between 2018 and 2023. Patient demographics, treatment duration, adverse events, and clinical outcomes were analyzed. **Results**: Data from 35 patients and 40 treatment episodes were analyzed, including 13 episodes in patients with HM, of whom 65.0% were male, with a median age of 69.0 years (IQR: 24.5 years). The median treatment duration was 13.5 days (IQR: 46.8), with a maximum of 203 days. TZD was switched from other anti-MRSA agents in 82.5% of cases, including 42.5% from LZD. One patient discontinued TZD due to liver dysfunction, attributed to concomitant medication use. Clinical cure rates were significantly higher in the non-HM group compared to the HM group (88.9% vs. 38.5%). The 90-day mortality rate differed notably between the HM and non-HM groups (69.2% and 3.7%). Despite 100% microbiological eradication, infection-related mortality rates were 3.7% in the non-HM and 38.5% in the HM group. No reported cases of optic neuritis, *Clostridioides difficile* colitis, or major bleeding; **Conclusions**: TZD appears to be safe for long-term use, regardless of HM status, with no major complications observed in this cohort.

## 1. Introduction

Methicillin-resistant *Staphylococcus aureus* (MRSA) is one of the most common pathogens that causes healthcare-associated infections. Although the prevalence of MRSA in hospitalized patients has been gradually decreasing, it remains a major drug-resistant pathogen isolated in hospital settings [1]. Various anti-MRSA antibiotics, such as tedizolid (TZD), vancomycin, teicoplanin, linezolid (LZD), and daptomycin, are approved for the treatment of hospital-acquired MRSA infections. TZD, a novel oxazolidinone antibiotic, was first developed in 2008 and approved in the United States in 2014 for the treatment of acute bacterial skin and skin structure infections caused by Gram-positive bacteria, including MRSA. In 2018, it was introduced in Japan as a new anti-MRSA drug for the treatment of skin and soft tissue infections (SSTIs). The pharmacological mechanism of TZD is similar to that of LZD; it binds to the 50S subunit of bacterial ribosomes and inhibits the formation of the 70S initiation complex, thereby suppressing protein synthesis and bacterial growth [2]. Tedizolid is a novel oxazolidinone antibiotic with linear pharmacokinetics and excellent tissue penetration. It demonstrates in vitro activity against a broad spectrum of Gram-positive bacteria, and it is four times more potent than linezolid, showing potential against linezolid-resistant pathogens. In neutropenic mouse models, bacteriostasis was achieved at high fAUC/MIC values, with granulocytes augmenting its antibacterial effects [3]. Several randomized controlled trials (RCTs) on SSTIs have demonstrated that the clinical cure rate of TZD is comparable to that of LZD [4,5,6,7]. LZD is known to cause bone marrow suppression, including thrombocytopenia and anemia, particularly when administered for >14 days or in patients with renal impairment [8,9]. Other reported adverse events associated with LZD include peripheral neuropathy and lactic acidosis, which are thought to be related to the inhibition of mitochondrial protein synthesis. The frequency of these adverse events is dose- and duration-dependent [10,11,12]. In contrast, TZD has been reported to cause fewer incidents of thrombocytopenia and gastrointestinal symptoms than LZD in RCTs [4,6,7]. The duration of TZD administration in these trials was 21 days. However, for conditions such as MRSA-associated chronic osteomyelitis, septic pulmonary embolism, deep abscesses, foreign body infections, or nontuberculous mycobacterial infections, treatment may require several months, making it challenging to continue LZD therapy due to its adverse effects. In such cases, the long-term use of TZD may be necessary. In clinical practice, febrile neutropenia guidelines recommend anti-MRSA agents for patients with hematologic malignancies [13]. In patients with hematological malignancies (HMs) experiencing febrile neutropenia (FN), the addition of anti-MRSA agents may be considered, with vancomycin recommended as the first-line choice. However, daptomycin and LZD have occasionally been used [14]. In patients with hematologic malignancies who have thrombocytopenia, impaired renal function, and require MRSA pneumonia coverage, administering vancomycin, daptomycin, and LZD may be challenging. In such cases, although off-label, TZD can be considered an alternative, for example, in ventilator-associated pneumonia, where Gram-positive cocci are detected, especially since it has been shown to be non-inferior to LZD in phase 3 trials [15]. Since the restriction on the duration of TZD administration was lifted in June 2019, allowing for long-term use, clinical cases of switching from LZD to TZD have been reported [16]. Nevertheless, the safety of long-term TZD use remains to be fully evaluated. Although the use of TZD is expanding, its safety and efficacy in patients with HM, particularly in terms of the risk of thrombocytopenia and bleeding in those with a low platelet count, have not yet been fully assessed. This study aims to evaluate the safety of TZD use based on clinical cases at our institution.

## 2. Results

### 2.1. Patient Selection and Grouping

Figure 1 shows the patient selection flow diagram. Initially, 47 episodes of TZD treatment were identified in 41 patients. After applying the exclusion criteria, 40 treatment episodes of 35 patients were included in the analysis. The exclusion criteria were as follows: non-Japanese patients (n = 5), a treatment duration of less than 3 days (n = 1), and duplicate patient data that were merged (n = 1). The remaining 35 patients were categorized into two groups based on the presence of HM. The non-HM group included 27 treatment episodes in 24 patients, while the HM group included 13 treatment episodes in 11 patients.

### 2.2. Baseline Characteristics

Table 1 shows the baseline characteristics of patients receiving TZD. The median age of the patients was 69.0 years (interquartile range [IQR]: 24.5). This study included 26 male patients (65.0%). The median duration of TZD administration was 13.5 days (IQR: 46.8), with a range of 4 to 203 days. A total of 19 treatment episodes (47.5%) involved administration for 14 days or longer, and 15 episodes (37.5%) involved administration for 28 days or longer. TZD was used as a switch from other anti-MRSA antibiotics in 33 patients (82.5%); in 17 patients (42.5%), the switch was specifically from LZD. Additionally, concurrent selective serotonin reuptake inhibitor use was observed in one case (2.50%), liver disease was present in nine cases (22.5%), and 19 cases (47.5%) had heart failure. Appendix A presents the detailed distribution of underlying diseases in patients with hepatic dysfunction, heart failure, and HM.

### 2.3. Laboratory Data at Admission

Appendix A shows the laboratory data of the patients receiving TZD at the time of admission. The median white blood cell count (WBC) was 5.25 × 10^3^/μL (IQR: 4.98 × 10^3^), with four cases (10.0%) presenting WBC levels of ≤500/μL. The median hemoglobin (Hgb) level was 9.75 g/dL (IQR: 3.75), with four patients (10.0%) showing Hgb levels of ≤7.0 g/dL. The median platelet count (PLT) was 12.9 × 10^4^/μL (IQR: 19.4), and 10 patients (25.0%) had a PLT count of ≤50,000/μL.

### 2.4. Sites of Infection and Pathogens

Table 2 summarizes the infection sites and corresponding pathogens identified in patients treated with TZD. The most common sites of infection were SSTIs and subcutaneous and intramuscular abscesses, accounting for eight cases (20.0%). Osteomyelitis was observed in six cases (15.0%), while vertebral osteomyelitis and septic arthritis were reported in four cases (10.0%). Other sites included pneumonia (seven cases, 17.5%), surgical site infections (SSIs) (five cases, 12.5%), and FN (four cases, 10.0%). Intravascular infections, such as catheter-related bloodstream infections (CRBSIs), pacemaker-lead infections, and thrombophlebitis, were noted in four cases (10%).

Regarding pathogens, there were 28 cases of documented infections and 14 cases of bacteremia. MRSA was the predominant pathogen, accounting for twenty episodes (50.0%), with seven episodes of MRSA bacteremia (17.5%). Methicillin-susceptible coagulase-negative Staphylococci were identified in four episodes (10.0%), all involving bacteremia.

### 2.5. Laboratory Data at the End of Treatment and Follow-Up

Appendix A shows the laboratory data at the end of TZD treatment. The median WBC count was 5.55 × 10^3^/μL (IQR: 4.90 × 10^3^), with 2.5% of patients having WBCs ≤ 500/μL. The median PLT count was 14.2 × 10^4^/μL (IQR: 17.1), with 22.5% (nine) of patients showing PLTs ≤ 50,000/μL. No cases of lactic acidosis were observed.

Appendix A shows the laboratory data obtained at the last follow-up of the patients receiving TZD. No abnormalities were observed.

### 2.6. Treatment Outcomes

Table 3 shows the outcomes at the end of TZD treatment.

Clinical cure was achieved in 72.5% (29/40) of cases, and microbiological eradication in 100% (12/12). The 30-day mortality rate was 20.0% (8/40), while the 90-day mortality rate was 25.0% (10/40), with infection-related mortality accounting for 15.0% (6/40) of deaths. No major adverse events were reported.

### 2.7. Platelet Count Transitions and Treatment Discontinuation

All patients described in Appendix A, which provides details on cases with platelet count transitions (from ≥50,000/μL before administration to ≤50,000/μL after administration), were patients with HM.

Appendix A outlines cases where TZD treatment was discontinued. One case involved treatment discontinuation due to liver dysfunction; however, this was considered unlikely to be primarily caused by TZD due to concomitant medication use. Appendix A details the 14 patients who received TZD treatment for more than 28 days.

Transfusions were performed in three cases, with one death reported. However, no cases involved the discontinuation of TZD, and the deaths were attributed to causes unrelated to infection.

### 2.8. Comparative Analysis: HM vs. Non-HM Groups

The baseline characteristics, laboratory data, outcomes, infection sites, and pathogens were analyzed by dividing the cases into two groups: non-HM and HM (Table 4, Table 5 and Table 6, Appendix A).

Patients with HM had higher rates of heart failure, liver dysfunction, and malignancy compared to non-HM patients. However, the duration of TZD administration was shorter in the HM group (Table 4).

Hemoglobin levels and platelet counts were lower at baseline, at the end of TZD treatment, and at the end of follow-up in the HM group (Appendix A). The HM group also had significantly higher severity, as indicated by intensive care unit admission rates and mechanical ventilation use, leading to a higher frequency of red blood cell and platelet transfusions, as well as higher mortality, including infection-related mortality (Table 5).

TZD was frequently used to treat FN in the HM group; however, the detection rate of MRSA was lower than that in the non-HM group (Table 6).

## 3. Discussion

We analyzed the cases of TZD administration in patients with hematologic and non-HM. This study included 35 Japanese patients and 40 treatment episodes, of which 11 patients (13 episodes) had HM. The median treatment was 13.5 days (maximum: 203 days), with a median of 19.0 days for non-HM patients and 12.0 days for patients with HM. Among the patients with HMs, 100% switched from other anti-MRSA agents, with 53.8% specifically switching from LZD.

The only reason for treatment discontinuation was liver dysfunction, which was considered unlikely to be primarily caused by TZD due to concomitant medication use. No cases of major bleeding were observed. MRSA infections and bacteremia were reported in twenty (50.0%) and seven (17.5%) episodes, respectively. There were 30 episodes of SSTIs and bone or joint infections; however, TZD was also used for other conditions, such as pneumonia and FN. No major complications were observed during the follow-up period, suggesting that TZD can be safely used not only in patients without HM but also in those with HMs.

A retrospective study from the United States involving 37 patients reported a median treatment duration of 188 days (IQR: 62–493 days), with 54.1% of patients receiving concomitant serotonergic agents [17]. The underlying conditions included prosthetic joint infection (29.7%), device-related spinal infection (18.9%), and osteomyelitis (13.5%). Adverse effects leading to treatment discontinuation included dizziness, lactic acidosis, and macrocytic anemia, each reported in 8.1% of the cases, with the treatment duration until discontinuation being 675, 745, and 123 days, respectively. There were no cases of peripheral neuropathy, optic neuritis, visual changes, or onset or the worsening of serotonin syndrome.

In a retrospective study of 81 patients receiving TZD in Spain, the median treatment duration was 28 days (IQR: 14–58 days). A total of 44.4% of patients had prior exposure to LZD. The most common indications for TZD use were prosthetic joint infections, acute bacterial SSTIs, and osteomyelitis. The reported adverse events included gastrointestinal disorders (2.5%), anemia (1.2%), and thrombocytopenia (7.4%). The median time to thrombocytopenia was 26.5 days (range, 17–58.5 days) [18].

A report from Japan described a retrospective study of eight cases in which patients with spondylitis who developed thrombocytopenia due to LZD were switched to TZD. The causative pathogens were MRSA in seven cases and *Enterococcus faecium* in one case, with six cases presenting with bacteremia. TZD treatment (mean duration: 30.3 ± 9.5 days) resulted in successful outcomes in all cases, with no treatment discontinuations due to thrombocytopenia or other adverse effects [16].

In our Japanese cohort, the sample size of 40 episodes was relatively large, with a median TZD treatment duration of 13.5 days and a maximum of 203 days. No lactic acidosis was observed, and treatment was discontinued in only one case due to liver dysfunction. However, since the patient was taking concomitant medications, this was not definitively attributed to TZD. Therefore, TZD is likely safe for long-term use.

There are few reports on the use of TZD in FN among patients with HM, although studies on LZD are more common. In RCTs and retrospective studies involving LZD for FN, no reports have indicated that it is inferior to vancomycin in terms of antipyretic effects or mortality rates, suggesting that LZD can be used effectively and safely [19,20].

In this study, transfusions were frequently performed in patients with HM; however, this could be influenced by the underlying disease, making it difficult to attribute this effect solely to TZD. Additionally, no adverse events, such as gastrointestinal bleeding, were observed, and there were no cases in which treatment was discontinued.

Although TZD is an off-label treatment, it is often chosen in situations where renal impairment makes vancomycin difficult to use, pneumonia cannot be ruled out, or daptomycin is unsuitable, leaving no other viable treatment options.

While the use of TZD for VAP is off-label, trials have shown that it is not inferior to LZD in terms of clinical response. Therefore, its use should be considered [15].

This was a retrospective analysis with a limited sample size and inherent bias. Moreover, there were a few cases in which TZD was used as an initial therapy, with most cases involving a switch from LZD, making it challenging to fully evaluate adverse effects. Furthermore, in patients with HM, many cases are severe, and TZD was often used in situations where disease progression likely worsens the patient’s condition. As this is a retrospective study, it inherently has limitations in capturing rare or long-term complications, as the follow-up period for some patients was short. While we made efforts to collect as much data as possible from available medical records, the retrospective nature of this study means that the underreporting of adverse events cannot be completely ruled out.

Since the analysis was based on treatment episodes rather than the first episode per patient, while efficacy could be assessed, the evaluation of adverse events might be overrepresented for the same patient. However, in this study, the number of patients reporting adverse events was small, so we consider this bias to be minimal.

Despite these limitations, this study represents a large-scale investigation of TZD use in Japanese patients with HM. Prospective studies and safety evaluations using real-world data from patients with HM are required.

## 4. Materials and Methods

### 4.1. Study Design

This retrospective cohort study was conducted at St. Luke’s International Hospital, Tokyo, Japan, from August 2018 to December 2023. This study included adult patients aged 15 years or older who received TZD treatment. For off-label use, clinicians provided a thorough explanation of TZD use to the patient and their family, and consent was obtained. This study was approved by the Ethics Committee of St. Luke’s International Hospital (approval number 23-R123). In this study, the cohorts were defined as follows: the non-HM cohort included patients with non-hematologic malignancies and inactive hematologic malignancies. The HM cohort consisted exclusively of patients with active hematologic malignancies.

### 4.2. Outcomes

The primary endpoint was the frequency of adverse events. Secondary endpoints included incidence rates of bone marrow suppression, peripheral neuropathy, optic neuropathy, lactic acidosis, pseudomembranous colitis, gastrointestinal symptoms, treatment failure rates, and mortality.

### 4.3. Data Collection

Information regarding the diagnosis of infections, patient demographics, vital signs, laboratory data at the time of admission, drugs used in combination with TZD, and comorbidities were collected. All data were extracted from the electronic medical records.

### 4.4. Statistical Analyses

Baseline characteristics, clinical outcomes, and adverse events in patients receiving TZD are summarized. Subgroup analysis using univariate methods was conducted to compare outcomes between the HM and non-HM groups. The bivariate analysis employed Chi-square (χ^2^) and Fisher’s exact tests for categorical variables and the Mann–Whitney U test for continuous variables. Differences were considered statistically significant at *p* < 0.05. All statistical analyses were performed using SPSS version 29.0 (IBM, Armonk, NY, USA).

## 5. Conclusions

Due to its efficacy and safety for long-term use, TZD could be an alternative to LZD in both non-HM and HM groups.

## Figures and Tables

**Figure 1 antibiotics-13-01237-f001:**
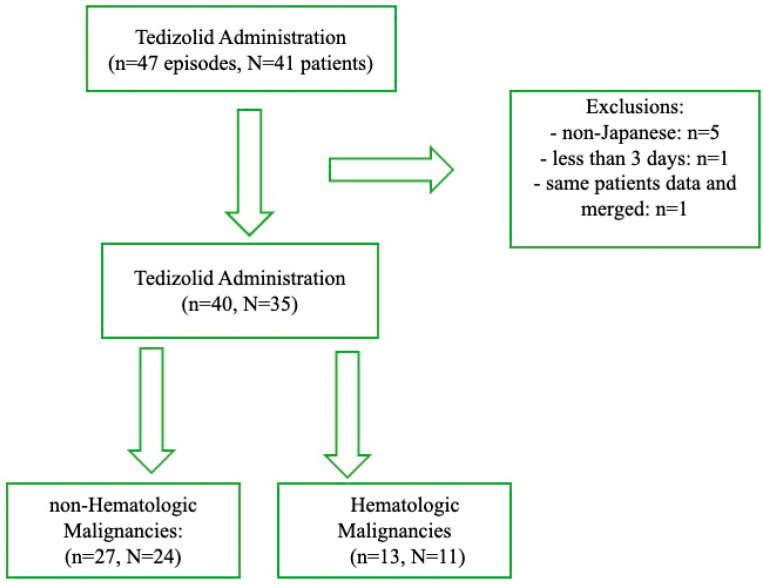
The diagram of patient selection flow.

**Table 1 antibiotics-13-01237-t001:** The baseline characteristics of the patients receiving TZD.

	n = 40 Episodes
Median Age (years) (IQR)	69.0 (24.5)
Male, n (%)	26 (65.0)
Median BMI (kg/m^2^) (IQR)	23.1 (6.31)
Median Duration of TZD Administration (day) (IQR)	13.5 (46.8) [4–203]
[Min–Max] ≥ 14 days, n (%), ≥28 days, n (%)	19 (47.5)/15 (37.5)
Switch from anti-MRSA antibiotics, n (%)	33 (82.5)
Switch from LZD, n (%)	17 (42.5)
Concurrent SSRI use, n (%)	1 (2.50)
Hypertension, n (%)	29 (72.5)
Heart failure, n (%)	19 (47.5)
Diabetes, n (%)	22 (55.0)
COPD/HIV, n (%)	0 (0.00)/0 (0.00)
Myocardial infarction, n (%)	2 (5.00)
Collagen disease, n (%)	1 (2.50)
Liver cirrhosis/Liver disease, n (%)	9 (22.5)
Malignancy, n (%)	32 (80.0)
Chronic renal failure, n (%)/Dialysis, n (%)	15 (37.5)/6 (15.0)
Cerebrovascular disease, n (%)	6 (15.0)

Abbreviations: IQR, interquartile range; MRSA: methicillin-resistant *Staphylococcus aureus*; BMI: body mass index; COPD: chronic obstructive pulmonary disease; SSRI: selective serotonin reuptake inhibitor; HIV: human immunodeficiency virus; and LZD: linezolid.

**Table 2 antibiotics-13-01237-t002:** Site of infection and pathogen detected in this study.

Site of Infection	n = 40 Episodes
Septic Arthritis, n (%)	4 (10.0)
Vertebral Osteomyelitis, n (%)	4 (10.0)
Osteomyelitis, n (%)	6 (15.0)
SSTI/Subcutaneous and Intramuscular Abscess, n (%)	8 (20.0)/8 (20.0)
Intravascular Infection, n (%)	4 (10.0)
CRBSI, n (%)	2 (5.00)
Pacemaker Lead Infections + Prosthetic Vascular Infections, n (%)	1 (2.50)
Thrombophlebitis, n (%)	1 (2.50)
Pneumoniae, n (%)	7 (17.5)
FN, n (%)	4 (10.0)
SSI, n (%)	5 (12.5)
Nosocomial Meningitis/CNS Infection/Epidural Abscess, n (%)	1 (2.50)/1 (2.50)/1 (2.50)
Pathogen	
Documented infection/bacteremia	28 (70.0)/14 (35.0)
MRSA infection/MRSA bacteremia, n (%)	20 (50.0)/7 (17.5)
MRCNS infection/MRCNS bacteremia, n (%)	4 (10.0)/4 (10.0)
*E. faecium* infection/*E. faecium* bacteremia, n (%)	2 (5.00)/2 (5.00)
*Corynebacterium* sp. infection/*Corynebacterium* sp. bacteremia, n (%)	2 (5.00)/1 (2.50)

Abbreviations: MRSA: methicillin-resistant *Staphylococcus aureus*; SSI: surgical site infection; SSTI: skin and soft tissue infection; CRBSI: catheter-related bloodstream infection; FN: febrile neutropenia; CNS: central nervous system; and MRCNS: methicillin-resistant coagulase-negative Staphylococci.

**Table 3 antibiotics-13-01237-t003:** The outcomes at the end of TZD treatment.

	n = 40 Episodes
Clinical cure, n (%)	29 (72.5)
Microbiological eradication, n (%)	12 (100)
30-day mortality, n (%)	8 (20.0)
90-day mortality, n (%)	10 (25.0)
Infection-related mortality, n (%)	6 (15.0)
ICU admission, n (%)	7 (17.5)
Mechanical ventilation	5 (12.5)
Vasopressor use, n (%)	7 (17.5)
Tedizolid allergy, n (%)	0 (0.00)
Endophthalmitis, n (%)	0 (0.00)
Clostridioides difficile colitis, n (%)	0 (0.00)
Gastrointestinal bleeding, n (%)	0 (0.00)
Cerebral hemorrhage, n (%)	0 (0.00)
RCC transfusion, n (%)	9 (22.5)
PLT transfusion, n (%)	6 (15.0)
G-CSF use, n (%)	4 (10)
Treatment discontinuation, n (%)	1 (2.50) (hepatic dysfunction)

Abbreviations: ICU, intensive care unit; RCC: red cell concentrate; PLT: platelet; and G-CSF: granulocyte-colony stimulating factor.

**Table 4 antibiotics-13-01237-t004:** The baseline characteristics of the patients receiving TZD between non-hematologic malignancies and hematologic malignancies.

	Non-Hematologic Malignancies: (n = 27)	Hematologic Malignancies: (n = 13)	*p*
Median age (year) (IQR)	72.0 (22.0)	69.0 (27.0)	0.716
Male, n (%)	17 (63.0)	9 (69.2)	1
Median BMI (kg/m^2^) (IQR)	23.2 (5.10)	21.1 (8.60)	0.93
Median days of Tedizolid administration (IQR) [Minimum–Maximum]	19.0 (66.0) [4–203]	12.0 (9.00) [4–28]	0.001
Switch from other anti-MRSA agents, n (%)	20 (74.1)	13 (100)	0.074
Switch from Linezolid, n (%)	9 (33.3)	7 (53.8)	0.631
Hypertension, n (%)	17 (63.0)	12 (92.3)	0.068
Cardiac failure, n (%)	8 (29.6)	11 (84.6)	0.001
Diabetes, n (%)	12 (44.4)	10 (76.9)	0.111
COPD/HIV, n (%)	0 (0.00)	0 (0.00)	
Myocardial infarction, n (%)	2 (7.40)	0 (0.00)	1
Connective tissue disease, n (%)	1 (3.70)	0 (0.00)	1
Liver cirrhosis/Liver disease, n (%)	3 (11.1)	6 (46.2)	0.038
Malignancy, n (%)	19 (70.4)	13 (100.0)	0.037
Chronic renal failure, n (%)	11 (40.7)	4 (30.8)	0.73
Dialysis, n (%)	3 (11.1)	3 (23.1)	0.37
Cerebrovascular disease, n (%)	3 (11.1)	3 (23.1)	0.37

Abbreviations: IQR: interquartile range; BMI: body mass index (kg/m^2^); MRSA: methicillin-resistant *Staphylococcus aureus*; COPD: chronic obstructive pulmonary disease; and HIV: human immunodeficiency virus.

**Table 5 antibiotics-13-01237-t005:** The outcomes at the end of TZD treatment between non-hematologic malignancies and hematologic malignancies.

	Non-Hematologic Malignancies: (n = 27)	Hematologic Malignancies: (n = 13)	*p*
Clinical cure, n (%)	24 (88.9)	5 (38.5)	0.003
Microbiological eradication, n (%)	5/5 (100)	7/7 (100)	
30-day mortality, n (%)	1 (3.70)	7 (53.8)	<0.01
90-day mortality, n (%)	1 (3.70)	9 (69.2)	<0.01
Infection-related mortality	1 (3.70)	5 (38.5)	
ICU admission, n (%)	0 (0.00)	7 (53.8)	<0.01
Mechanical ventilation, n (%)	0 (0.00)	5 (38.5)	0.002
Vasopressor use, n (%)	0 (0.00)	7 (53.8)	<0.01
RCC transfusion, n (%)	3 (11.1)	6 (46.2)	0.038
PLT transfusion, n (%)	2 (7.41)	4 (30.8)	0.075
G-CSF use, n (%)	0 (0.0)	4 (30.8)	0.008

Abbreviations: ICU: intensive care unit; RCC: red cell concentrate; PLT: platelet; and G-CSF: granulocyte-colony stimulating factor.

**Table 6 antibiotics-13-01237-t006:** Site of infection and pathogen detected in this study between non-hematologic malignancies and hematologic malignancies.

Site of Infection	Non-Hematologic Malignancies: (n = 27)	Hematologic Malignancies: (n = 13)
Septic Arthritis, n (%)	3 (11.1)	1 (7.69)
Vertebral Osteomyelitis, n (%)	4 (14.8)	0 (0.00)
Osteomyelitis, n (%)	6 (22.2)	0 (0.00)
SSTI/Subcutaneous and Intramuscular Abscess, n (%)	12 (44.4)	2 (15.4)
Intravascular Infection, n (%)	3 (11.1)	11 (84.6)
CRBSI, n (%)	2 (7.41)	0 (0.00)
Pacemaker Lead Infections + Prosthetic Vascular Infections, n (%)	1 (3.70)	0 (0.00)
Thrombophlebitis, n (%)	0 (0.00)	1 (7.69)
Pneumoniae, n (%)	6 (22.2)	1 (7.69)
FN, n (%)	0 (0.00)	4 (30.8)
SSI, n (%)	4 (14.8)	1 (7.69)
Nosocomial Meningitis/CNS Infection/Epidural Abscess, n (%)	0 (0.00)	1 (7.69)
Pathogen	Non-Hematologic Malignancies: (n = 27)	Hematologic Malignancies: (n = 13)
MRSA infection/MRSA bacteremia, n (%)	17 (63.0)/7 (25.9)	3 (23.0)/0 (0.00)
MRCNS infection/MRCNS bacteremia, n (%)	2 (7.41)/2 (7.41)	2 (15.4)/2 (15.4)
*E. faecium* infection/*E. faecium* bacteremia, n (%)	0 (0.00)/0 (0.00)	2 (15.4)/1 (7.69)
*Corynebacterium* sp. infection/*Corynebacterium* sp. bacteremia, n (%)	2 (7.41)/0 (0.00)	2 (15.4)/2 (15.4)

Abbreviations: MRSA: methicillin-resistant *Staphylococcus aureus*; SSI: surgical site infection; SSTI: skin and soft tissue infection; CRBSI: catheter-related bloodstream infection; FN: febrile neutropenia; CNS: central nervous system; and MRCNS: methicillin-resistant coagulase-negative Staphylococci.

## Data Availability

Due to the nature of this research and to protect participants’ privacy, the data were not shared publicly; therefore, supporting data are not available.

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
