# Peer review of "Study on the Efficacy and Safety of Tedizolid in Japanese Patients"

_antibiotics, 2024, doi:10.3390/antibiotics13121237_

Round 1

Reviewer 1 Report

Comments and Suggestions for Authors

Dear Authors,

Your research aims to add information to the use of tedizolid. it is important for clinical practice. However, the reason for the old references you used is not clear. There are recent articles dealing with the pharmacokinetic and pharmacodynamic characteristics of tedizolid. I also ask for clarification on the treatment of febrile neutropenia and thrombocytopenia in cases where they have been recorded.

Sincerely,

Reviewer 2 Report

Comments and Suggestions for Authors

This study evaluates the safety and efficacy of long-term tedizolid use in Japanese patients, including those with hematological malignancies, by retrospectively analyzing medical records. The results suggest that TZD is safe and effective for extended durations, even in high-risk patient groups, with minimal adverse events. A few issues need to be addressed before being considered for acceptance.

1. The study includes 40 treatment episodes in 35 patients, which may limit generalizability. Are these findings representative of broader patient populations, especially outside Japan?

2. The study relies on retrospective data, which is prone to bias and confounding factors. How do the authors address these limitations, and would a prospective study yield more robust conclusions?

3. The HM group had significantly poorer outcomes compared to non-HM patients (e.g., higher mortality rates). Could underlying conditions or disease severity in this subgroup confound the results?

4. Although no major adverse events were reported, the sample size and short follow-up for some patients may underestimate rare or long-term complications. How does the study address potential underreporting?

5. The study suggests TZD as an alternative to linezolid (LZD) but provides limited direct comparative data. How does TZD's cost-effectiveness, safety, and efficacy compare to LZD or other alternatives in similar clinical scenarios?

Reviewer 3 Report

Comments and Suggestions for Authors

The manuscript (ID: antibiotics-3312785) describes an analysis of the efficacy and safety of tedizolid in adult Japanese patients, including those with hematological malignancies (HM), indicating its favorable efficacy and safety. As the long-term use of linezolid is limited by hematological side effects, a study analyzing the long-term use of tedizolid in patients with HM is welcome.  

The retrospective cohort study is well designed and has relatively large sample size. In my opinion, the article is suitable for publication, but requires some modifications.

Comments:

-        Not all patients enrolled to cohort „non-Hematologic Malignancies“ had a malignancy (other than HM). Exactly 19/27 or 70% of the non-HM patients had a malignancy. Therefore, this patient cohort should be named differently throughout the paper.

-        Page 4, line 124 (title of Table 2): It is unnecessary to write „in our cases“ .

-        Table 2: The abbreviation SSI is not explained

-        The abbreviation MCNS is used in Table 2, but he other abbreviation (MRCNS) is explained.

-        Page 5, lines 142-144 Please check and correct this: The number of cases with clinical cure is 25/36 (69.4%) as stated in the text, or 29 (72,5%) as stated in Table 3? The microbiological eradication is 11/11 (text) or 12/12 (Table 3)? Infection-related mortality is 60% (text) or  6% (Table 3)?

-        The titles of Table 4 and Table 5 should be changed: „group“ or „cohort“ should be added at the end of the title. The cohort of Non-Hematologic Malignancies should be renamed. Not all patients enrolled to cohort "non-Hematologic Malignancies" had a malignancy (other than HM). Precisely 19/27 or 70% of the non-HM patients had a malignancy. Therefore, this cohort of patients should be named differently throughout the paper.

-        Since episodes of tedizolid treatment are analysed instead of patients, a selection bias is possible which should be explained in the discussion.

-       -  Page 8, line 190: „site“ should be capitalised „Site“

 -        The title of Table 6 should be corrected in line with the suggestions for Tables 4 and 5. It is also unnecesary to write „in our casas“ in the title.

 -        Table 6: The abbreviation SSI is not explained.

-        - The abbreviation MCNS is used in Table 6, but the other abbreviation MRCNS is explained.

- Page 9, lines 198-199: Please correct the sentence according to the previous suggestion - The cohort of Non-Hematologic Malignancies should be renamed.

Page 9, line 229 – bracket after MRSA – tipfeler?

Page 12, line 341: “ 9Gerson“ should be corrected
